# Imaging of Hepatitis B Virus Nucleic Acids: Current Advances and Challenges

**DOI:** 10.3390/v14030557

**Published:** 2022-03-08

**Authors:** Luisa F. Bustamante-Jaramillo, Joshua Fingal, Marie-Lise Blondot, Gustaf E. Rydell, Michael Kann

**Affiliations:** 1Department of Infectious Diseases, Institute of Biomedicine, Sahlgrenska Academy, University of Gothenburg, 405 30 Gothenburg, Sweden; luisa.fernanda.bustamante.jaramillo@gu.se (L.F.B.-J.); joshua.fingal@gu.se (J.F.); gustaf.rydell@gu.se (G.E.R.); 2Microbiologie Fondamentale et Pathogénicité (MFP), CNRS UMR 5234, University of Bordeaux, 33076 Bordeaux, France; marie-lise.blondot@u-bordeaux.fr; 3Region Västra Götaland, Department of Clinical Microbiology, Sahlgrenska University Hospital, 405 30 Gothenburg, Sweden

**Keywords:** hepatitis B virus, cccDNA, RNA transcripts, imaging, single cell analysis, in situ hybridization (ISH), fluorescent in situ hybridization (FISH), branched chain DNA (bDNA), sandwich nucleic acid hybridization, CRISPR/cas9, MS2, PP7, Sun-Tag, aptamer, molecular beacon, quantum dots, click chemistry, peptide nucleic acids (PNA), ClampFISH, anchor, OR protein

## Abstract

Hepatitis B virus infections are the main reason for hepatocellular carcinoma development. Current treatment reduces the viral load but rarely leads to virus elimination. Despite its medical importance, little is known about infection dynamics on the cellular level not at least due to technical obstacles. Regardless of infections leading to extreme viral loads, which may reach 10^10^ virions per mL serum, hepatitis B viruses are of low abundance and productivity in individual cells. Imaging of the infections in cells is thus a particular challenge especially for cccDNA that exists only in a few copies. The review describes the significance of microscopical approaches on genome and transcript detection for understanding hepatitis B virus infections, implications for understanding treatment outcomes, and recent microscopical approaches, which have not been applied in HBV research.

## 1. Problems and Significance of cccDNA and mRNA Visualisation

With more than 850,000 deaths per year, hepatitis B virus (HBV)-infections are a major global health burden [1]. The high death-toll is mainly caused by chronic infections as they can lead to liver cirrhosis and to primary hepatocellular carcinoma (HCC). Treatments by nucleotide/nucleoside analogues (e.g., tenofovir disoproxil, adefovir and entecavir), which inhibit the HBV (and HIV) reverse transcriptase [2], efficiently decrease the hepadnaviral load with minimal side effects. Although disease progression and HCC risk are reduced, albeit only after several years of treatment, current treatments are unsatisfactory as viral clearance is rare [1]. Accordingly, new treatment options are currently under investigation.

cccDNA—HBV has a partially double-stranded DNA genome (relaxed circular; rc) and infects hepatocytes. Infection leads to a nuclear form of the HBV genome, which is a covalently closed circular (ccc) episomal molecule [3]. cccDNA is the template for progeny viral genomes, which are synthesized via reverse transcription of a pregenomic RNA (pgRNA) and subsequent second strand DNA synthesis within the viral capsid [3]. HBV is considered to be a non-cytotoxic stealth virus [4] and intranuclear cccDNA is thought to persist throughout the cells’ long lifespan, which, in healthy conditions is estimated at around 200–400 days in rodents [3,5]. The fraction of infected hepatocytes varies with the viral load, ranging between <1% and nearly 100%, as shown by immune histology detecting the HBV core protein in chimpanzees with acute HBV infection [6,7]. Current treatments by nucleos(t)ide analogues (NA) block rcDNA synthesis by inhibiting the viral polymerase, which mediates reverse transcription and second strand synthesis but does not affect the cccDNA template. That is why is clearance of chronic HBV infections under treatment is never reached [8,9], and so therapeutic interventions targeting cccDNA would represent the “holy grail”. 

As cccDNA represents the HBV reservoir, understanding the mechanisms which underly its persistence remains an important issue. At the beginning of HBV chronic infection, cccDNA molecules become transcriptionally active resulting in detectable levels of the HBe antigen. In later infection during seroconversion, there is a loss of the HBe antigen which is associated with a more efficient immune response against infected hepatocytes and, consequently, a reduction in the cccDNA reservoir [10]. The high rate of hepatocyte cell death will induce liver regeneration through mitosis. Because of its specific structure as a minichromosome lacking a centromere, cccDNA molecules should be distributed asymmetrically to daughter cells [11]. This is supported by the study conducted by Allweiss et al., where a strong cccDNA reduction was shown in liver of human chimeric mice after in vivo proliferation of HBV-infected primary human hepatocytes (PHH) [12]. A sole study has suggested cccDNA survival upon mitosis in ducks infected with duck HBV (DHBV). The differences in these findings could be explained by species specificity between DHBV and human HBV regarding the formation and amplification of the cccDNA pool mechanisms in hepatocytes [13]. Recently, Lythgoe et al. developed a novel mathematical model based on published data to infer the lifespan of cccDNA [14]. However, their estimations alone could not explain the long durations of chronic infection observed in patients treated with NA. These findings support the hypothesis suggested by Alweiss and others that either a sub-population of quiescent hepatocytes harbouring cccDNA molecules persists during chronic infection or that NA therapy does not suppress all viral replication. This could be an explanation of why HBV clearance is never reached in chronically infected patients.

A number of phenomena require a better understanding of cccDNA dynamics [15]. While the reduction in the cccDNA load by the elimination of infected hepatocytes is evident, cccDNA may also be reduced by non-cytopathic T cell effector mechanisms [16,17,18]. The underlying mechanisms of cccDNA dynamics remain largely elusive, which includes cccDNA’s fate upon cell division. As liver regeneration does not involve stem cells [19], daughter cells are derived from potentially infected differentiated hepatocytes. The viral genomes may thus become incorporated into the nuclei of the daughter cells which would lead to persistence exceeding the lifespan of the hepatocyte. Alternatively, the viral genomes may be passively trapped into the nuclei of the daughter cells, which would—due to the ratio of cytosolic and nuclear space—result in a dilution of 1:7. Yet unsolved is the question of whether cytoplasmic cccDNA molecules remain stealth or if they are recognized by the host’s innate immune response. In view of potential drugs targeting cccDNA, it remains unknown whether all cells eliminate cccDNA or if cccDNA molecules in a subset of cells escape treatment. 

The analysis of cccDNA by visualising techniques on a single-cell level is hampered by a number of problems. (I) The copy number is low, which mostly ranges between one to four molecules per infected hepatocyte [20]. This makes it difficult to obtain convincing signal-to-noise ratios. (II) Infected cells contain cytosolic capsids which harbour replication intermediates with complete HBV (−)-strand DNA and partial (+)-strand DNA. While their cytosolic location enables discrimination from the strictly nuclear cccDNA, direct evidence is difficult due to the similarity of these two DNA forms. Selective detection of cccDNA might be obtained by adding sensors with double-strand specificity. (III) Already early in infection, cells contain HBV DNA integrated into hosts’ chromatin [21]. Although often rearranged and fragmented, this DNA comprises the same DNA sequence as cccDNA in a linear form (Table 1) [22].

The obstacles in visualising specific HBV DNA molecules led to the use of surrogate markers to estimate the proportion of infected cells as, e.g., the expression of viral proteins, namely, the core (HBcAg) and surface (HBsAg) antigens. Early studies in 1980s added in situ hybridizations (ISH) to immune histological stains, yet most non-histological publications nowadays are restricted to the average of cccDNA molecules found in numerous cells by qPCR.

HBV transcripts—Similar to cccDNA quantifications, most current studies describe the average of HBV transcripts in multiple cells, which limits the information of the experiments. This became evident by droplet digital PCR experiments on multiple pieces of explant tissue, which indicated large variations of viral transcript copy numbers and of cccDNA within the liver, as well as large variations in the ratio between different transcripts [25]. Furthermore, single-cell RNA sequencing data derived from HBV-infected HuH7.5-NTCP cells indicate a wide cell-dependent range though of low abundance (74% of infected cells: >15 copies, 26%: <15 copies; based on 360.000 mRNA copies per cell) [26]. These numbers were within range of those found in liver biopsies of five HBV-infected patients by single-cell laser capture microdissection and droplet digital PCR, in which mRNA/cccDNA ratios between 4 and 533 were observed [20]. Of note, four of the five patients’ hepatocytes with cccDNA did not have detectable HBV transcripts, which indicates that knowledge about HBV DNA and mRNA in individual cells is important when transcription-silencing treatment approaches shall be introduced. 

HBV RNA detection by visualisation techniques is challenging, but in contrast to cccDNA it does not require the melting of double-stranded structures (see ISH). Nonetheless, the detection of specific transcripts is impeded by their similarities (Table 1). HBV transcripts are 3′ co-terminal and differ just in their 5′ extension. Furthermore, splice variants which contribute to replication and pathogenesis are described and are biomarkers in HBV infection [27]. The low copy numbers—in contrast to highly abundant cellular RNAs—imply that the crowding of signals beyond the limit of microscopy is unlikely given the current resolution limits of microscopy (lateral resolution: approx. 120 × 120 nm last generation confocal microscopy, wave-length-dependent; 60 × 60 nm sparse deconvolution during real-time microscopy [28], 20 × 20 nm stimulated emission depletion (STED) microscopy; axial resolution approx. 3-fold). 

The aim of this review is to provide an overview of the challenges of HBV genome- and transcript-detection (Table 1), and how microscopical studies increased the understanding of HBV infection (Table 2). Furthermore, recent approaches which have not been applied to HBV research are described and their potential for single-molecule detection and tracking of HBV nucleic acids are discussed (Table 3). 

## 2. Techniques Applied for HBV Nucleic Acid Visualisation

### 2.1. Electron Microscopy

The low number of cccDNA molecules argue against the use of electron microscopy. Consequently, investigations on cccDNA structure by electron microscopy were performed after enrichment and purification by gradient centrifugation. Pioneering work by Newbold et al. showed that the cccDNA of DHBV exists in a heterogeneous population [29], and Bock et al. showed that HBV cccDNA is organized in nucleosomes associated with histones and the viral core protein [30]. 

### 2.2. In Situ Hybridisation

ISH is based on the binding of a labelled probe to a complementary sequence (Figure 1). Fixation increases the cells’ permeability for probe diffusion to the detection site. When double-stranded sequences are targeted, deproteinization by pronase/proteinase K is applied, though it is not mandatory (Figure 1A), and a melting step is required which in turn needs sample fixation (Figure 1C). For co-localisation with proteins after deproteinization, the images of two sequential experiments must be merged. Antigen detection by (mostly) immune fluorescence is applied, followed by the hybridization step. While low resolution images, e.g., detection in specific cell compartments, can be easily achieved, high-resolution images in the few hundred nm range are far more complex, as the mechanical and thermal manipulations change the morphology. Nonetheless, a precision of around 100 nm can be obtained as demonstrated by co-localisation of the adenovirus 5 genome with the attached adenoviral protein VII [66].

Another obstacle is the specificity of the probe in conjunction with the hybridization conditions. For single-stranded nucleic acid detection, probes of opposite orientation are convenient controls, but they can evidently not be applied when cccDNA shall be monitored. Further specificity controls may be based on scrambled sequence, mutual sequences, or cells in which the target sequences are absent.

## 3. Principal Findings on HBV by Nucleic Acid Visualisation

### 3.1. Tissue Samples—Imaging with Low Resolution

The first detection of HBV DNA with ISH in infected human liver tissue was performed on sections using radioactive probes generated by nick translation from a plasmid [31]. The technique was used in a number of subsequent studies with probes labelled with ^3^H, ^35^S, or ^125^I [32,33,34,35,67] and was adjusted to be compatible with formalin-fixed, paraffin-embedded liver sections [36,68]. Later, biotin-labelled probes were used together with an avidine–peroxidase complex [37,38], as well as digoxigenin-labelled probes applied with alkaline phosphatase-conjugated antibodies directed against digoxigenin [39]. The studies identified the cytoplasm as the site where most HBV DNA localise and exhibited two distinct patterns, either diffused staining in the majority of hepatocytes or staining confined to foci [69]. By treating liver sections with RNase and using ISH without denaturation, probes which target the single-stranded region of rcDNA predominantly showed staining in the cytoplasm [35]. Conversely, weaker nuclear signals which were only obtained after denaturation and were resistant to RNase treatment suggested the presence of cccDNA or integrated HBV DNA [34,70]. 

Co-localisation studies of HBV nucleic acids and HBV antigens by Nidobitek et al., showed that HBV nucleic acids are present in HBV antigen-positive cells only [71], but hybridization signals were also obtained in cells which were HBcAg-negative but HbsAg-positive. In today’s view, this finding indicates transcription from integrated HBV DNA, as classical integrates are linearized between core promotor and core ORF [22], even though differences in the sensitivity between antibodies towards HBcAg and HBsAg could also be a contributing factor. It has been suggested that HBsAg regulates cccDNA levels with high levels of HBsAg production blocking cccDNA synthesis [72]. 

Later, branched amplification technique originally designed for RNA detection and adapted to DNA (branched chain DNA; bDNA; sandwich nucleic acid hybridization method) were used (Figure 2). The principle of this signal amplification technique is that two primary oligonucleotides (target probes), to which a non-hybridizing sequence is attached, bind to the target sequence at adjacent sites (Figure 2A). In a second step, so-called pre-amplifier oligonucleotides hybridize to both non-target sequence-bound parts of the first oligonucleotides when they are in proximity (Figure 2B), thus increasing the specificity of detection. The oligonucleotide is attached to a redundant sequence to which multiple oligonucleotides (amplifier molecules) with another extension bind, forming a tree-like structure in which the extensions are the “leaves” (Figure 2C). Labelled probes binding to the extensions of the amplifier are then added in a third step (Figure 2D), which allows amplification by 8000-fold compared to ISH using a direct single probe. The method thus allows the detection of single molecules. Different extensions on the primary oligonucleotides allow the detection of two to four target sequences in parallel. Thus, several probe sets may be used to allow specific detection of nucleic acids with only partial accessibility or degradations.

Zhang et al. [46] applied this amplification system (View RNA ISH technology) using NBT/BCIP-labelled probes (nitro blue tetrazolium/5-bromo-4-chloro-3-indolyl-phosphate), which were visualised by the alkaline phosphatase-mediated colour change of a dye. HBV RNA, DNA, and cccDNA were detected in 160 fixed biopsies from chronically HBV-infected patients. Three primary oligonucleotides were used in this study. The first probe detects the (+)-strand sequence found on pgRNA, cccDNA, and rcDNA; the second targets the (-)-strand found in cccDNA and rcDNA; the third is directed against the (+)-strand DNA absent in rcDNA. Specificity for DNA and RNA was achieved by treatments with RNase A/RNase H- and DNase, respectively. Further discrimination between rcDNA and cccDNA was obtained by their cytosolic or nuclear localisation. The fixed samples were dewaxed and rehydrated before immune detection of viral antigens, which was followed by ISH after deproteinization. They observed that (-)-strand HBV DNA in the cytoplasm and nucleus though cytosolic HBV DNA was only observed in sample from patients with high viral loads (>1.6 × 10^7^ mL serum). An exclusive nuclear stain was found at low viral loads (<1 × 10^4^ mL serum), which was independent from the presence of HBeAg in patients’ serum. The analysis of samples from patients after one year of adefovir treatment exhibited the expected absence of cytoplasmic HBV DNA but persistence of nuclear DNA. Considering that no discrimination of integrated HBV DNA and cccDNA was included, the persistent nuclear stain could be explained by integrates instead of cccDNA, which is in agreement with the strong correlation of HBsAg positivity.

In a follow-up study, the group analysed biopsies from 313 patients by bDNA detection [47]. Here, HBV DNA abundance strongly correlated with serum viral load (r = 0.68), HBsAg (r = 0.53), and HBcrAg, which is a composite of antigens bearing different HBc epitopes. This includes HBeAg, which is assumed to be a surrogate serum marker for cccDNA in HBeAg-positive patients (r = 0.65). Higher intrahepatic DNA correlated with fibrosis and inflammation, indicating that ISH-derived signals reflect not only intrahepatic replication but also disease progression. The correlation between HBV DNA ISH signal and serum HBV DNA load was confirmed in a follow-up study with liver biopsies from 94 patients with chronic HBV and 10 control patients [48]. The distribution of intrahepatic HBV DNA was found to show a spatial relationship with collagen fibres supporting that the assay can provide information about disease progression. Further findings using bDNA were published by Wang et al., where the RNA Scope bDNA technology was used over the time-consuming ViewRNA ISH technology. DNase-treated tissue sections from entecavir-treated patients were studied, in which it was found that HBV RNA–positive hepatocytes were clustered in foci distributed across the lobules and co-localised with HBV surface antigen [43].

### 3.2. Single Cell Analyses—Towards High-Resolution Imaging

Detecting HBV DNA and RNA by autoradiography or enzymatic labelling allows analyses of high cell numbers, but these techniques are limited by their low resolution. The resolution limits were overcome with the introduction of fluorescent probes. Fluorescence microscopy was developed in the early 1900 by Zeiss and Reichert, first published in two papers in 1911 [73,74], but it would be another 70 years until in situ fluorescence detection of nucleic acids (FISH) was available [75,76]. 

In HBV research, the high-resolution detection of DNA and RNA by FISH has mainly been applied for analyses of hepatoma cell lines and primary cells, though later research also showed its successful use in stem-cell-derived hepatocytes and humanized or chimeric mice [44,45,77,78]. Using FITC-labelled RNA probes of different polarity directed against the single-stranded part of rcDNA in hepatoma cell lines, lipofected with exogenous capsids, Rabe et al. could show that HBV genome liberation occurs at the nuclear envelope or inside the nucleus [40]. However, as capsid-lipofection of cells yields hundreds of intranuclear HBV genome copies, this technique did not require high sensitivity as would be required for HBV cccDNA and rcDNA detection upon natural infection. The same technique was applied on digitonin-permeabilized cells to which capsids were added, showing that the capsids need the cellular nuclear import factors importin α and importin β for interaction with the nuclear pore. It was further deduced that genome release occurs at the inner face of the nuclear pore complex, which was supported by electron microscopy imaging [40,41].

In 2017, Zhang et al. [49] investigated the spatial distribution of HBV DNA and RNA in HepAD38 cells, which are stably transfected with cDNA corresponding to pgRNA, and HBV-infected HepG2-NTCP cells using fluorescent bDNA amplification (View RNA ISH technology) after deproteinization with probes directed against (−)- and (+)-strand HBV DNA. Stochastic Optical Reconstruction (super resolution) microscopy (STORM) not only determined the number of HBV molecules per cell but also the 3D reconstruction of their distribution. In contrast to classical FISH which resulted in rare signals, bDNA detection was much more efficient showing 8.2 (median; SD 12.2) (−)-strand and 4.3 (median; SD 6.6) (+)-strand molecules per HepG2-NTCP cell. These numbers are slightly higher than numbers reported from single-cell PCR quantification of patient samples. Rarely, overlapping intranuclear signals which indicate cccDNA were observed. As expected, the analysis of HepAD38 cells showed much higher numbers of (−)-strands and (+)-strands with medians of 110 (SD 36) and 53.8 (SD 27.5), respectively. Of note, HBV DNA signals showed a random distribution, though no proteins or cellular structures were visualised.

Fluorescence-based microscopy was also used in the parallel detection of HBV DNA and RNA with proteins. The combination allowed the co-localisation of HBV episomes with the viral HBxAg, which enhances HBV transcription, preferentially in nuclear domains, which are actively transcribed [79]. This result was achieved in infected HepaRG cells, detecting HBV DNA using 3′ digoxigenin-labelled oligonucleotides, which were subsequently visualised by Cy3-labelled anti-digoxigenin antibodies. A similar study on the localisation of transcriptionally inactive cccDNA of HBxAg-negative mutant virus was published by Tang et al. [42]. Using 57-nt-long oligonucleotides, labelled at their 5′ and 3′ ends with the bleaching-stable Alexa647 fluorescent dye on RNase-treated cells, the authors observed DNA episomes preferentially in the vicinity of chromosome 19. Co-localisation with structural maintenance of chromosomes 5/6 complex (SMC5/6), which represses transcription, revealed that transcriptionally active cccDNA re-localise randomly, at least in HepG2-NTCP cells. In contrast to other studies, dozens of cccDNA molecules, but mostly fewer than three, were observed, showing that PHH were more efficiently infected.

Li et al. investigated the quantity and the fate of cccDNA molecules in HBV- and DHBV-infected hepatoma cell lines [80]. FISH was applied using biotinylated plasmid DNA which was detected by fluorescein-labelled avidin, thus detecting both DNA strands. Specificity for DNA was obtained by RNase digestion of the samples prior to hybridization. HBV cccDNA copy numbers of up to 45 (median of eight) were revealed, well above in vivo findings. The results for DHBV showed up to 60 copies of cccDNA per cell (median 19). While in agreement with findings from infected primary duck hepatocytes [81], these numbers are significantly higher than those derived from single-cell analyses of DHBV-infected livers, which range between 2.9 and 8.6 copies per infected cell [82]. Of note, the assay was not specific for cccDNA but would also detect rcDNA from viral capsids that had entered the nucleus that is converted to cccDNA through protein free rcDNA. Following the cccDNA copy numbers after several rounds of cell divisions showed a decrease in the number per cell, and led to the conclusion of a symmetric distribution in DHBV. However, a symmetrical passage to daughter cells was not confirmed for HBV, where a loss of cccDNA was rather observed.

In an extensive recent work, Yue et al. analysed the spatial resolution of proteins and HBV DNA and RNA in hepatoma cells [50]. They used the ViewRNA ISH technology with strand-specific Cy3- and Cy5-labelled probes for DNA/RNA detection and immune fluorescence for protein detection, avoiding deproteinization. Furthermore, they detected ribosomes by adding labelled puromycin, which is incorporated into the C-terminus of elongated nascent protein chains, thereby terminating translation. Plus-strand DNA and pgRNA was distinguished by DNase digestion. Analysing cells at different time points after infection, they observed up to 6 (-)-strand DNA molecules per cell after 15 days post-infection. pgRNA copy numbers increased from one to a maximal nine (median six) during that period. The target region used to detect pgRNA is also found on precore mRNA (encoding HBeAg), but RT-qPCR quantification with primers specific to precore mRNA suggested significantly lower concentrations than pgRNA. cccDNA was visualised by a probe which targets the region of the plus (+)-strand that is missing in rcDNA and was shown to co-localise with acetylated histone 3 and RNA polymerase II. 

Adding entecavir, which blocks reverse transcription, reduced the number of DNA molecules but not pgRNA. Furthermore, treatment with Interferon α reduced both DNA and RNA, while inhibitors of capsid assembly reduced the number of pgRNA puncta co-localised with capsid by 48 to 57%. Super-resolution microscopy by STED further showed that nuclear HBV DNA molecules, as detected by the (+)-strand DNA, co-localised with acetylated histone 3 (H3K27ac) and RNA polymerase II. Quantifying pgRNA, capsid- and ribosome co-localisation revealed a ratio of 3.9 capsid-localised pgRNA: 1 free pregenome, and that rapid translation is followed by slow encapsidation and HBV genome maturation. They further showed that microtubules are needed for virion secretion by promoting multivesicular body formation.

In summary, several studies showed that detection of HBV DNA and transcripts are possible, although the specific forms of DNA and RNA remained frequently unclear (Table 2). Zhang et al. [46] and Yue et al. [50] showed that cccDNA can be discriminated from rcDNA by probe systems targeting the region of the (+)- strand that is absent from rcDNA. However, this target region is also found on integrated HBV DNA, which therefore cannot be distinguished from cccDNA by this approach [46]. The fact that the HBV transcripts are 3′ co-terminal and differ just in their 5′ extension complicates specific detection. Efforts to circumvent this problem include quantification using multiple target regions and then calculating the amount of each transcript, as well as more sophisticated methods [83,84,85,86]. Nonetheless, the microscopical data broadened the knowledge on molecular intracellular events on the single-cell level, albeit no time-lapse experiments unravelled the kinetics of HBV DNA and RNA.

## 4. Recent Developments in DNA and mRNA Detection for Living Cells

Extrapolating the technical developments leading to more sensitive sensors and more sophisticated software for optimizing signal-to-noise ratios will improve the detection of rare signals for HBV DNA and RNA detection. While these developments are difficult to anticipate, there are novel techniques which could possibly allow time-lapse-based kinetics which have not yet been applied to HBV research (Table 3). Due to the complexity of HBV imaging, it is likely that not a single method but a hybrid of methods could allow accurate discrimination of nucleic acids in cells. In this section, methods which have previously shown promise in viral nucleic acid visualisation are briefly discussed, as well as their potential for single-molecule detection and tracking of nucleic acids.

### 4.1. CRISPR-Based Detection

The CRISPR system is based on the primitive bacterial immune response to viral infections [87]. CRISPR RNAs (crRNAs), trans-activating crRNA (tracrRNA), and Cas proteins form a ribonucleoprotein complex capable of crRNA-guided recognition and degradation of target nucleic acids. Single-guide RNA (sgRNA) combines the crRNA and tracrRNA into a single RNA molecule. crRNA contains 17–20 nucleotides complementary to the target DNA, while tracrRNA serves as scaffold for the Cas nuclease. The specificity is determined by the sequence of the sgRNA, which hybridises to the target DNA and is recognised by Cas in the presence of a DNA protospacer-adjacent motif (PAM) in the complementary region close to the target site [88,89].

The specificity of CRISPR/Cas9 for HBV was demonstrated in an experimental therapeutic attempt [90]. However, the imaging of HBV nucleic acids by CRSPR/Cas has not been published yet but might be promising. This can be achieved by using catalytically inactive Cas (dCas9; D10A/H840A) that bind specifically to DNA sequences without target cleavage [89,91]. In analogy with FISH, site-specific detection requires specific signal amplification for visualisation. Zhou et al. described a couple of different approaches that could be used for that [92]. For instance, Chen et al. [93] used enhanced green fluorescent protein (EGFP)-tagged dCas9 and a structurally optimized sgRNA to image repetitive elements and non-repetitive genomic sequences. However, in order to obtain sufficient signal strength, at least 26 sgRNA targeting the same gene had to be used for detection of the non-repetitive genomic elements. Other strategies to improve the signal strength were used by Deng et al. [94], which allowed the detection of sequences by dCas9/sgRNA complex without denaturation in fixed cells, though this strategy was more successful for repetitive sequences. Qin et al. [95] re-engineered sgRNA with MS2 motifs derived from bacteriophage MS2 that can bind to the MS2-coat protein (MCP). Fused to a fluorescent protein, this approach allowed the detection of non-repetitive elements using just four sgRNA. Later, Fu et al. [96] added RNA stem-loop motifs of MS2 and PP7 (from bacteriophage PP7) to sgRNA in order to bind MCP and PP7-coat protein (PCP) proteins, allowing the parallel detection of two sequences. 

Another version of CRISPR named CRISPRainbow was developed to follow up to six loci in living cells [97]. The authors designed an improved version of CRISPRainbow, called the CRISPR-Sirius system, by increasing the sgRNA stability and brightness of low-copy genomic loci by introducing MS2 and PP7 aptamers in the sgRNA tetraloop [98]. Chaudhary et al. [52] engineered a new CRISPR-based technology, which allowed detection by decreasing background signals with high target specificity, even for the visualisation of genome loci with small number of sequence repeats or RNA (Figure 3). This technology combines tri-partite fluorescent proteins (sfGFP) with the Sun-Tag system in living cells [52] (Figure 3). The Sun-Tag system is based on a scaffold protein that contains 24 copies of short epitopes GCN4 (General Control protein 4), which is recognized by the scFv (Single Chain Fragment variables) antibody fused to GFP, which is expressed from a separate vector [53]. This system amplifies the fluorescent signal, allowing the detection of single molecule in living cells. Another approach that may be applied to detect HBV dsDNA is SensiTive Recognition of Individual DNA Ends (STRIDE), which relies on CRISPR/Cas9 capacity to induce DNA breaks (double-stranded cleavages or single-strand nicks, Figure 4A). The 3’ ends of the breaks are then elongated by a polymerase adding modified nucleotides (Figure 4B), which are recognised by two different primary antibodies that bind in close proximity (Figure 4C). The antibodies are then detected by secondary antibodies conjugated to oligonucleotide (Figure 4D). These oligonucleotides bind to another oligonucleotide, which becomes circular (Figure 4E), allowing the elongation of the template oligonucleotide by a DNA polymerase in a rolling circle (Figure 4F). The nascent single strand DNA is then visualised by fluorescent oligonucleotide (Figure 4G) [23]. 

Single-stranded RNA (ssRNA) is also recognized by Cas9 with high affinity, matching the Cas9-associated guide RNA sequence when the PAM is presented in trans as a separate DNA oligonucleotide (PAMmers) [99]. Thus, dCas9 has been also used for RNA visualisation, but the challenge of improving the amplification of the signal in low-abundance RNA remains. This might be overcome by using the so-called CRISPR Sunspot, which allows signal amplification sufficient for single mRNA molecule detection in living cells (Figure 5) [24]. The system requires different elements. (I) A stable cell line expressing dCas9-24 × GCN_v4, under control of the inducible TRE3G promotor, which was then transduced to obtain scFv-sfGFP proteins with nuclear localisation signal, also expressed under inducible TRE3G promoter. As described before, the 24 × GCN_v4 tags allowed signal amplification by recruitment of the scFv-sfGFP proteins. (II) The signal from a single-molecule mRNA was amplified using a plasmid containing two sets of three different sgRNA for targeting three sites of the single mRNA molecule. Finally, the stable cell line was co-transfected with the sgRNA and PAMmers vectors [24]. Attempts at using other dCas molecules as, e.g., dCas13 for endogenous RNA imaging, however, showed worse signal-to-noise ratios than other CRISPR approaches [100]. 

Advantages and Drawbacks of CRISPR in HBV DNA/RNA Visualisation.

Advantages: The CRISPR system can be used in fixed but also living cells, allowing kinetic studies with single-molecule detection. In HBV analyses, a single-stranded cleavage Cas can be used to differentiate partial dsDNA from cccDNA. Additionally, different Cas can be combined to detect DNA and RNA, although the problems of signal-to-noise ratios must be addressed. A clear advantage of CRISPR-based strategies is that they do not require modification of the viral genome, thus allowing the analysis of wt HBV, but the systems require the expression of specific factors in the host cells.

Drawbacks: As described above, sufficient signal amplification is a major obstacle in the application of CRISPR/cas-based systems. The CRISPR Sunspot system seem to overcome this restriction but the use of large complexes such as Sun-tag may interfere with the kinetic of nucleic acids. 

### 4.2. Non-CRISPR-Based Techniques

#### 4.2.1. RNA Imaging

Aptamer systems typically require two main components: the expression of (chimeric) mRNA fused to an aptamer sequence which forms a stem-loop, and an RNA-binding fluorescent partner [54]. When co-expressed, the fluorescent molecule binds to the aptamer’s structure motif resulting in foci (Figure 6A). Aptamer systems which utilize fluorescently-tagged proteins such as MS2-MS2 coat protein (MS2-MCP) have been applied for the visualisation of RNA molecules in living cells. Furthermore, these aptamer systems have been used in conjunction with CRISPR-based techniques for imaging in living cells [98]. However, these aptamer–protein systems have a high background due to the continuous expression of fluorescent proteins and may thus not be suitable for single-molecule detection of nucleic acids.

To improve the background-to-signal ratio, light-up aptamers such as Spinach, Spinach2, and RNA Mango were developed. Conditional fluorophores are utilized which only fold into a fluorescent form upon aptamer binding, which makes these systems relatively background-free (Figure 6B) [55]. The most commonly used light-up aptamer systems, Spinach and its improved version, Spinach2, contain two helical segments flanking a G-quadruplex (G4) structural motif, which serves as a platform for the binding of conditional GFP-like fluorophores such as 3,5-difluoro-4-hydroxybenzylidene imidazolinone (DFHBI) [101]. Spinach2 has been used to visualise the cell-to-cell transfer of Sindbis RNA in BHK cells [102] and HIV RNAs in living 293T cells, albeit with reproducibly weak signal [103]. Although the photostability of fluorophores and aptamer-folding conditions have hindered single-molecule sensitivity detection, recent improvements in these factors have made the live-cell tracking of single mRNAs and non-coding RNAs possible with RNA Mango II arrays [56]. Such a system could potentially be used for single-molecule tracking of HBV transcripts (and pgRNA), though additional probes would be required for distinguishing the different molecules. A drawback to all aptamer systems is that they require genetic modification of the virus to include aptamer sequences.

Another method commonly used for labelling of intracellular RNA are molecular beacons (MB) [57]. These nanodevices contain a target sequence flanked by self-complimentary palindromic sequences which form a stem-loop (Figure 7A) [57,58]. Typically, the 3′ and 5′ ends of the MB are conjugated with a fluorophore and quencher, respectively. Upon hybridisation with the target sequence, the fluorophore and quencher are distanced from each other, resulting in fluorescence emission (Figure 7B). An advantage over aptamer-based labelling is that MBs do not require the engineering of the virus and generally exhibit low noise [58], although imperfect quenching of the fluorophore or degradation by nucleases may still result in background fluorescence [57]. Furthermore, MBs are prone to non-specific protein binding, which can lead to non-specific signals. These limitations can be overcome by proper experimental conditions and the synthesis of MB backbones with unnatural nucleotide analogues which provide resistance to nucleases. [57]. Although MBs have shown their potential in the tracking of viral RNAs in living cells (see below), the delivery of MBs into living cells requires additional manipulations, which was achieved through methods such as microinjection and reversible permeabilization with streptolysin O [57,58]. 

Numerous viral RNAs have been visualised using MBs in living cells. Aspects of bovine respiratory syncytial virus pathogenesis such as cell-to-cell spreading was tracked in bovine turbinate cells [104]. While not demonstrated, the authors suggested that a confocal microscope with photon-counting capabilities could allow single-molecule detection. By utilizing multiple MBs to concentrate fluorescence, single Japanese encephalitis virus genomes could be tracked in living BHK-21 cells following their release from the capsid [105]. A modified MB strategy utilizing hairpin DNA-functionalized gold nanoparticles which enables cell entry and improves nuclease-resistance has been utilized for spatiotemporal studies of respiratory syncytial virus transcripts [106]. Additionally, coxsackievirus RNAs [107,108], poliovirus (+)-strand RNA [109], and Influenza A virus transcripts [110] have been successfully labelled with MB probes in living cells.

#### 4.2.2. dsDNA Imaging

The ANCHOR system has shown potential for tracking of viral genomes, which is based on the recognition of a specific double-stranded DNA sequence (ANCH3) by a bacterial protein (OR protein) fused with a fluorescent marker. OR binds cooperatively to the ANCH3 sequence, thereby also binding to flanking sequences. This leads to the accumulation of c. 400 molecules to the target and thus to a signal, which far exceeds the fluorescence from unbound fluorescent OR. The binding of OR is dynamic, having a half-life of around 40 sec. This replenishment of fluorescence on the target DNA has the advantage that bleached molecules become replaced allowing longer observation periods only limited by phototoxicity to the cell. 

This system has been already adapted to several viruses such HIV-1 [111], human cytomegalovirus [112], human adenovirus [113], and Vaccinia virus [114]. As this system requires double-stranded DNA, the insertion of the ANCH3 sequence in the single-stranded portion of rcDNA would result in foci only after repair to cccDNA. While adaptation to HBV is possible, it requires engineering of the virus and—as at least one HBV ORF is affected—expression by trans-complementation. This makes investigations of HBV spread difficult as trans-complementation is possible but at low efficiency [115]. Furthermore, the cell has to express the fluorescent OR protein, so potential impacts of OR on the host cell have to be excluded. 

Typical DNA- and RNA-based probes require a single-stranded molecule for their hybridization with a target sequence which complicates dsDNA detection. Peptide nucleic acids (PNA) are synthetic DNA-like molecules which contain a polyamide backbone in the place of deoxyribose phosphate and are unique in their ability to also hybridize with duplex sequences through various modes (Figure 8). The binding mode of PNA probes is dependent on its nucleobase’s composition. They hybridize with complimentary DNA or RNA sequences with high specificity and affinity due their uncharged polymer backbone [61]. Despite these unique features, the use of PNA-based probes for imaging of nucleic acids are limited likely due to challenges with cell entry and non-specific binding to hydrophobic surfaces [61,62]. However, these challenges can be overcome by making modifications to their backbone or through conjugation to charged or hydrophilic groups [62].

PNA probes have been used for the labelling of both single-stranded and double-stranded nucleic acids in living cells. For instance, PNA-based probes have been used to visualise genomic telomeres in live U2OS and HeLa cells [116] and multiplex imaging of Influenza H1N1 transcripts in living MDCK cells [117]. For the labelling of plasmid DNA, an N-succinimidyl-3-(2-pyridyldithio) propionate (SPDP)-PNA linker was used as it provides reactive sites for both dsDNA and quantum dots. By the transfection of labelled plasmids into CHO-K1 cells, transcriptionally active plasmids similar to cccDNA could be tracked on a single-molecule level [118].

Aside from direct labelling, PNAs may also have use for specific dsDNA labelling by locally exposing hybridization sites without sample denaturation for secondary probes (PNA openers). In a hybrid FISH-based method, a PNA opener allowed padlock-probes access to ssDNA for rolling circle amplification to visualise single copies of genomic DNA [119]. While not yet applicable for imaging, PNA openers have also been applied to MB strategies for the detection of specific dsDNA sequences [63].

Avoiding sample denaturation, the detection of DNA virus genomes has been achieved through metabolic labelling with ethynyl-modified nucleosides compatible with copper(I)-catalyzed azide-alkyne cycloaddition (“click”) chemistry [65]. Newly synthesized adenovirus, herpesvirus, and vaccinia virus genomes were successfully labelled and packaged into virions without interfering with infectivity. A combination with other methods such as MBs may allow specific detection of rcDNA. However, the detection of single modified genomes through signal amplification requires cell fixation, which means time-lapse microscopy cannot be applied.

Many attempts for the real-time imaging of viral genomes require direct binding to nucleic acid sequences or repeats, which often requires genetic modification of the virus or limits application to single-stranded sequences. An attractive option would be the visualisation of native viral genomes in situ through their structural properties such as the HBV pgRNA epsilon. Another potential target is the G4 structural motif, which has been recently identified in the pre-core promotor region of cccDNA through pulldown assays to the G4-binder DHX36 [120]. G4 structures, which form in GC-rich regions of RNA and DNA, have been identified in a variety of nucleic acids including untranslated regions of mRNAs, human telomeres, and genomes of viruses belonging to most Baltimore classes [121,122]. 

Lou et al. developed a “light up” benzothiazole-based fluorogenic probe which binds to the G4 structure of HCV RNA genomes with high specificity and low background in living cells [123], with the viral G4 essentially functioning as an aptamer. While the authors suggest that the high specificity may in part be due to viral G4s outnumbering host-cell G4s due to high replication, a guiding probe such as PNA could potentially broaden application to specific-binding to low abundance G4s. In another study, the G4-binding molecule NDI conjugated to a PNA probe was used to selectively target a dsDNA G4 in the HIV-1 LTR region [124]. In principle, the PNA probe targets a dsDNA sequence in vicinity of a G4 structure which allows the direct stacking of the G4-binder NDI to the structural motif. The intrinsic fluorescence of NDI coupled with the possibility to introduce cellular localisation signals to the conjugates would allow imaging of HIV-1 G4s and, indirectly, HIV-1 genomes in living cells. Selective targeting of the HBV G4 or other secondary structures may allow imaging of cccDNA and possibly distinguish from other HBV nucleic acids.

#### 4.2.3. Alternative Fluorophores

A general hurdle for tracking of nucleic acids in living cells are the photostability of organic fluorophores [59], as is the case for methods such as MB and aptamer systems. As an alternative, an increasing number of fluorescent nanomaterials are being developed due to their improved photostability, such as gold nanoparticles and quantum dots (QD) [125,126]. QD are fluorescent nanocrystals ideal for single-molecule imaging due to their photostability and strong signals. However, drawbacks which include cytotoxicity, their large size, and their non-specific binding have limited wider applications [60].

QD nanobeacons (QD-NB) have been utilized for single virus tracking through the labelling of different viral components [59]. A QB-NB conjugated with a black hole quencher and a single phosphorothioate co-modified DNA allowed the tracking of single HIV-1 RNA in HIV-1 integrated cells [127]. Furthermore, labelled HIV-1 genomes in progeny viruses allowed the tracking of genome release on a single virus level. In another study, modified genomic RNA of HIV-1 lentivirus labelled with QDs and encapsidated in vesicular stomatitis virus glycoproteins allowed for single virus tracking [128].

In this section, some techniques already applied for the visualisation of viral nuclei acids have been discussed. However, there are other techniques for the visualisation of nucleic acid, where DNA-binding proteins, such as zinc-fingers (ZFs) and transcription activator-like effectors (TALEs), have been used to for the direct imaging of genomic loci in living cells. The DNA-binding protein is fused with FPs and is an alternative to ISH techniques. Some DNA-binding proteins have high binding specificity for cognate DNA sequences, and DNA denaturation is not required [88].

## 5. Discussion

Microscopical imaging of HBV nucleic acids has contributed significantly to the understanding of the viral infection process. Developing from work performed with nucleic acid probes detected with radioactivity and chemical staining, followed by fluorescent probes and signal amplification using bDNA technology, *in situ* hybridization has broadened the knowledge on the intracellular trafficking of HBV RNA and DNA. The recent technical developments within microscopical nucleic acid analysis described in this review could increase this knowledge further. While these methods have not yet been shown to be compatible with HBV, they have a large potential to increase both detection sensitivity and specificity and live-cell tracking of HBV nucleic acids with time-lapse microscopy.

The labelling of HBV nucleic acids in living cells raise many other problems depending on the nature of the virus itself, the type of genome (DNA, RNA, single or double strand), the genome organization, the capsid size limits, the safety classification, and the model (tissue, cell culture, live-cell imaging). The intrinsic peculiarities of HBV genomes render the design and engineering of recombinant vectors technically highly challenging. For instance, insertion in the viral polymerase gene which comprises two-thirds of the viral genome would need to be compensated by a functional polymerase in trans. Nonetheless, studies have shown that viral transcription is far more efficient when initiated in cis by the viral polymerase [115]. The spacer region in the viral polymerase was demonstrated to be an interesting location for insertions, given that it is the least conserved and structured region [129,130]. Furthermore, intact HBx ORF seems to be required for a fully active recombinant HBV virus [131,132]. Careful attention needs to be paid to insertion sites, as they should not introduce stop codon given the overlapping HBV ORFs. From the perspectives of deciphering the HBV life cycle, a high number of recombinant particles is a pre-requisite. This relies not only on the recombinant vector but also on the use of cellular models that support infection such as Huh-7, HepG2 cells, or other liver cell lines expressing the NTCP receptor.

Considering all these parameters, CRISPR-Cas9 and ANCHOR technologies appear to be the best methods to study single HBV nucleic acids in living cells. Both strategies need expression of specific factors in the host cells to allow recognition of viral nucleic acids. In the case of the ANCHOR system, supplemental engineering is necessary to produce recombinant HBV virions containing the specific ANCH3 sequence to detect dsDNA. CRISPR-based systems, however, have the potential to both label RNAs and DNAs in living cells and could possibly enable the visualisation of nucleic acids from patient-derived HBV. Distinguishing similar HBV nucleic acids from another remain a central problem. One option could be to design sgRNAs to detect the different modalities of the virus genome. For example, a single-stranded cleavage Cas can be used to differentiate the partial rcDNA from cccDNA. In addition, different Cas can be combined to detect DNA and RNA. The combination of techniques could result in a promising way to address the imaging of HBV nuclei acids. CRISPR is a system compatible with different techniques such as aptamers or complexes such as Sun-Tag in the Sunspot system. However, nucleic acids are dynamic, making it difficult to capture the precise time of detection, and the use of large complexes such as Sunspot can interfere with their kinetics. This is one of the disadvantages of CRISPR Sun-tag. However, it is possible to use a smaller version of Cas9, its orthologue saCas9. Regarding the ANCHOR system, one major advantage is the dynamic association between the OR and ANCH3 sequence, which was demonstrated to not interfere with gene expression. 

The choice of the target sequence for CRISPR or the insertion site for ANCH3 have to be carefully considered but should theoretically be conceivable. We have great hope that in the future, such technologies will be developed and allow for answering difficult questions such as genome release, site of rcDNA repair to cccDNA, the potential existence of viral replication centres, and the fate of cccDNA upon cell division.

In summary, several new techniques have evolved, mostly allowing the detection of specific RNA and DNAs by time-lapse microscopy with improved signal-to-noise ratios. As they depend on host cell or viral genome modifications, they are limited in their use for patient-derived samples. However, the molecular knowledge they could bring to HBV nucleic acids and kinetics could prove invaluable to research towards new therapeutics. 

## Figures and Tables

**Figure 1 viruses-14-00557-f001:**
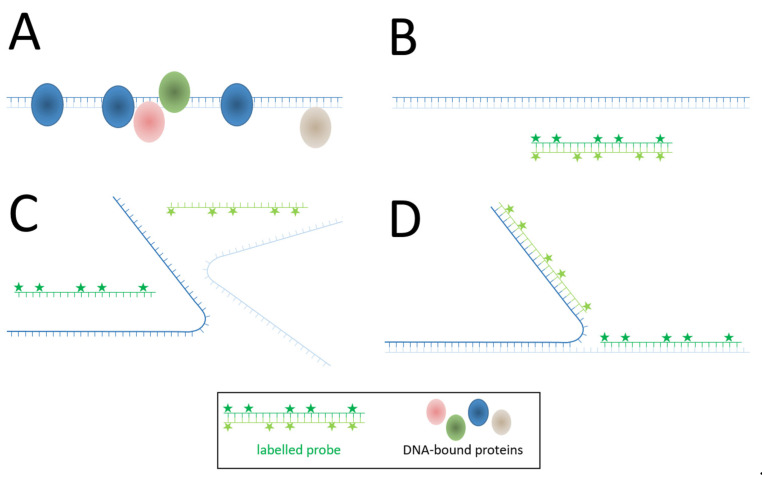
Classical in situ hybridization (ISH); (**A**) Deproteinization of dsDNA; (**B**) Addition of fluorophore-labelled dsDNA probe; (**C**) Melting of dsDNA and probe; (**D**) Hybridization. More information on the graphical elements is given in the box within the figure.

**Figure 2 viruses-14-00557-f002:**
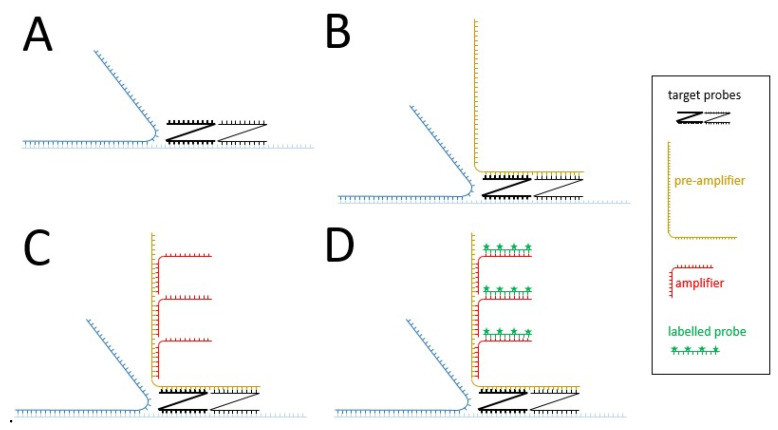
Branched chain DNA (bDNA) assay for detecting target sequences on dsDNA: (**A**) Hybridization of two different target probes to dsDNA after melting; (**B**) Hybridization of the pre-amplifier to the free nucleotides of the two adjacent target probes; (**C**) Hybridization of identical amplifier molecules to the pre-amplifier; (**D**) Binding of labelled probes to the amplifier molecules. More information on the graphical elements is given in the box within the figure.

**Figure 3 viruses-14-00557-f003:**
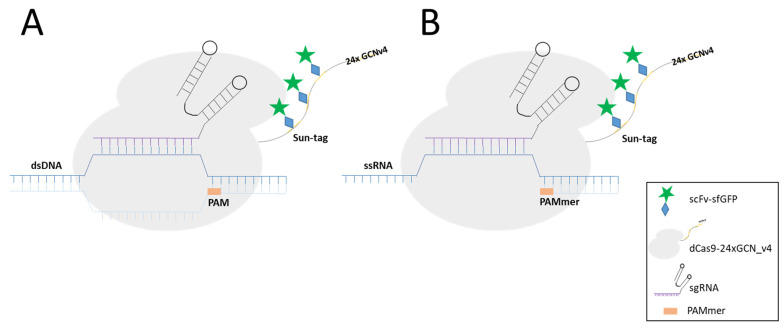
CRISPR/dCas9 in genome visualisation. (**A**) dsDNA detection. dCas9 binds to PAM and sgRNA. dCas9 is fused to a single fluorescent protein or conjugated to multiple fluorescent proteins through Sun-Tag. (**B**) RNA detection. As in A., but the PAM is extended by an oligonucleotide (PAMmer). More information on the graphical elements is given in the box within the figure.

**Figure 4 viruses-14-00557-f004:**
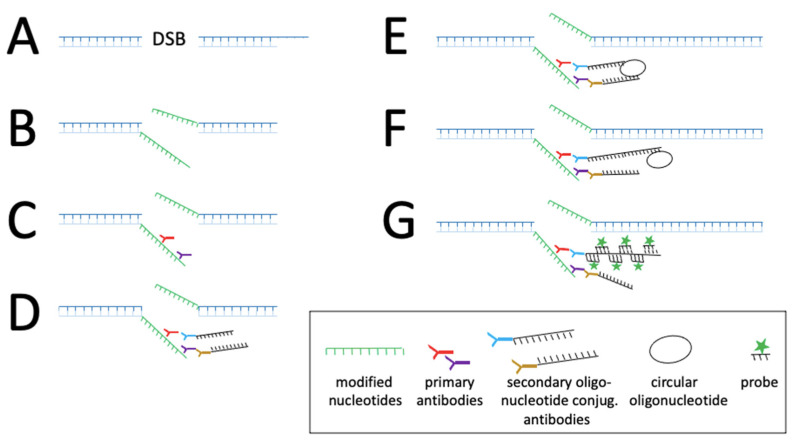
STRIDE for sequence detection on dsDNA: (**A**) CRISPR/Cas9-mediated induction of double-strand breaks (DSB); (**B**) Modified deoxynucleotides are enzymatically conjugated to 3′ DNA ends by Pol I; (**C**) Two different primary antibodies, directed against the modified nucleotides bind in close proximity; (**D**) Secondary antibodies conjugated with oligonucleotides bind to primary antibodies; (**E**) Another oligonucleotide binds to the two oligonucleotides from D., forming a circular DNA template; (**F**) One oligonucleotide from D. acts as primer for elongation in a rolling circle mechanism by a DNA polymerase using the circular DNA from E. as template thereby generating a ssDNA with repetitive motifs; (**G**) Hybridization of fluorophore-labelled oligonucleotides to the ssDNA from the rolling circle. More information on the graphical elements is given in the box within the figure.

**Figure 5 viruses-14-00557-f005:**
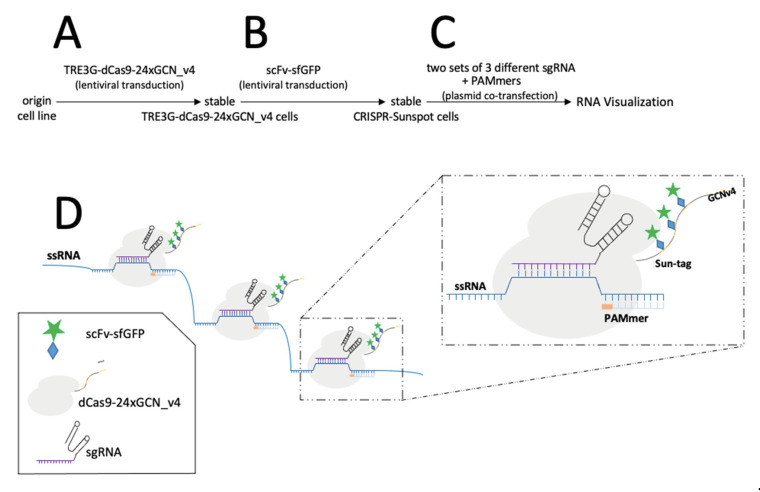
Simplified schematic presentation of CRISPR-Sunspot system: (**A**) Transduction of a cell line with a lentivirus expressing TRE3G-dCas9-24xGCN_V4; (**B**) The transduced cells are with a second lentivirus expressing scFv-sfGFP lentivirus; (**C**) Co-transfection of two plasmids of the double-transduced cells; the first coding for two sets of three different sgRNAs each, the second plasmid expresses the PAMmers; (**D**) Detection and visualisation of the target sequence. sgRNAs and PAMmer bind to the target sequence allowing recruitment of Sun-tagged Cas9. More information on the graphical elements is given in the box within the figure.

**Figure 6 viruses-14-00557-f006:**
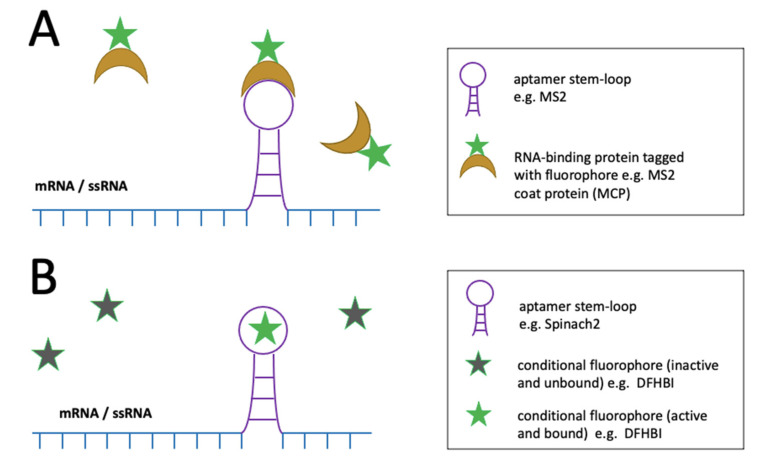
Schematic representation of aptamer systems for the detection of mRNA and ssRNA. The stem-loop formed (purple) by the aptamer sequence of chimeric mRNA provides a binding platform for fluorescently-tagged proteins in aptamer–protein systems (**A**) or conditional fluorophores in light-up aptamer systems (**B**). In light-up aptamer systems, fluorescence emission only occurs after aptamer-binding of the conditional fluorophore which makes these systems background-free. More information on the graphical elements is given in the boxes within the figure.

**Figure 7 viruses-14-00557-f007:**
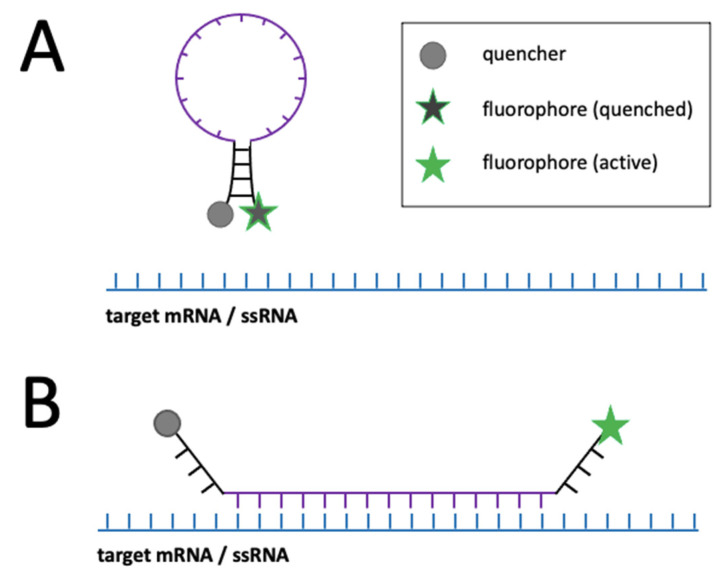
Schematic representation of mRNA detection with molecular beacons: (**A**) The self-complimentary sequence of the molecular beacon forms a stem (black), thereby keeping the fluorophore and fluorescence quencher in close vicinity effectively quenching the fluorophore; (**B**) Upon hybridization of the molecular beacon’s complimentary sequence (purple) with the target, the quencher is distanced from the fluorophore which consequently regains its fluorescent properties. More information on the graphical elements is given in the boxes within the figure.

**Figure 8 viruses-14-00557-f008:**
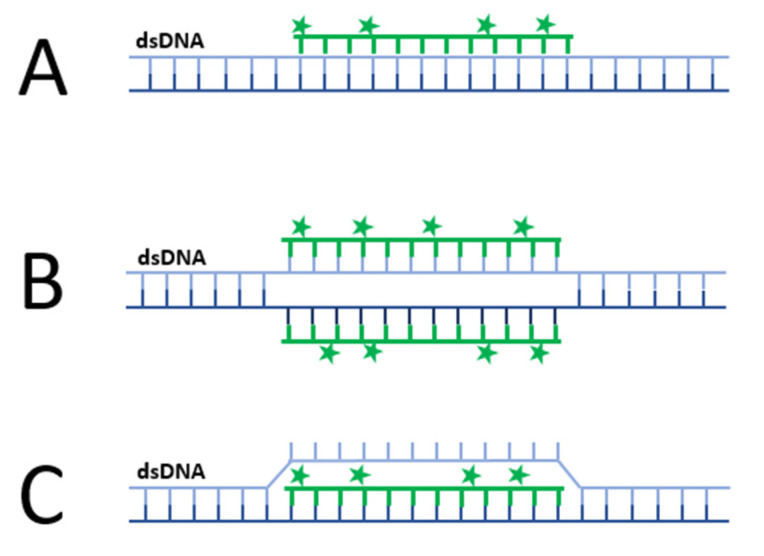
Schematic representation of the most common modes PNA-based probes hybridise with dsDNA: (**A**) Fluorescent PNA probes (green) can be designed to directly label dsDNA by forming a triple helical structure (triplex); (**B**) As in A., but by double duplex invasion; or (**C**) by duplex invasion, among other binding modes. Alternatively, as duplex invasion (**C**) leads to strand displacement, a PNA opener can make hybridization sites available for secondary probes without sample denaturation or a fixation step.

**Table 1 viruses-14-00557-t001:** Problems and possible solutions in HBV nucleic acid detection.

Nucleic Acid Form	Description/Encoded	Problems in the Detection	Possibilities to Overcome These Problems
DNA			
cccDNA (3.2 kb)	Covalently closed circular DNA Template for progeny viral genomes	(A) Cross reaction with rcDNA, (B) dslDNA, (C) integrated DNA comprises the same sequence than cccDNA but linear. (D) Low copy, high signal/noise ratio	(A) Target the region that is single-stranded in rcDNA. (B, C) Target the region in which the linearization typically occurred. (D) Single molecules detection, amplification of signal, sensors specific of double strand nucleic acids (e.g., CRISPR/STRIDE [23])
rcDNA (3.2 kb)	Relaxed circular DNA	Cross reaction with cccDNA, dslDNA and integrated DNA.	Make assay specific for single strand DNA
dslDNA (3.2 kb)	Double-stranded linear DNA	Same sequence as cccDNA but linear	
Integrated DNA (3.2 kb or shorter)		Same sequence as cccDNA but linear	
RNA		(A) Low-abundance in contrast with cellular mRNA, high background. (B) Similarity between transcripts: most transcripts from cccDNA use the same poly A site and thereby share the same 3′ end.	(A) Single molecules detection and amplification (e.g., CRISPR/Sunspot system [24]) (B) Discrimination between transcripts can be achieved by combining multiple target regions, e.g., hypothetically by using aptamers (see illustration below), beacons, or designing sgRNAs targeting multiple places in CRISPR system and comparing between the signal from each transcript.
PrecoreRNA (3.5 kb)	Involved in infection and propagation	(A) Cross reaction with other cccDNA derived transcripts. (B) Cross reaction with transcripts from integrated HBV DNA	(A) Combining multiples target (see above) (B) Transcripts from integrated HBV DNA typically start at the start of Pre S, S or X, lack the 3’ end including the typical HBV poly A site and are fusion transcripts using a host gene poly A site.
Pre-genomic RNA (3.5 kb)	Template for viral genome Core protein (capsid protein) Polymerase	(A) Cross reaction with other cccDNA derived transcripts (B) Cross reaction with transcripts from integrated HBV DNA.	e.g., 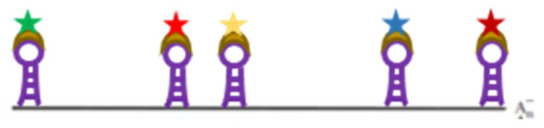
Pre S (2.4 kb)	L HBsAg (envelope protein)	(A) Cross reaction with other cccDNA derived transcripts. (B) Cross reaction with transcripts from integrated HBV DNA	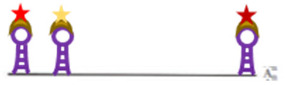
S RNA (2.1 kb)	S HBsAg, M HBsAg (envelope proteins)	(A) Cross reaction with other cccDNA derived transcripts. (B) Cross reaction with transcripts from integrated HBV DNA	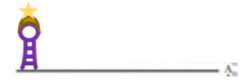
X RNA (0.7 kb)	Regulatory X protein	Cross reaction with other cccDNA derived transcripts (B) Cross reaction with transcripts from integrated HBV DNA.	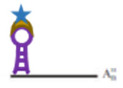
From integrations (S)		Cross reaction with cccDNA derived transcripts.	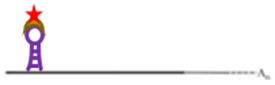

Symbols: see Figure 6.

**Table 2 viruses-14-00557-t002:** Advantages and disadvantages of techniques for HBV nucleic acid detection.

Technique	Modality	Target	Principal Findings	Advantages	Drawbacks	References
Electron microscopy		cccDNA	• cccDNA exist in heterogeneous population • cccDNA associated with nucleosomes and viral core protein	• High magnification and resolution	• Strong training • Expensive • Low number cccDNA requires enrichment and purification	[29,30]
In situ hybridisation	• Radioactive probes • Biotin • Digoxigenin RNase addition: -without denaturation -with denaturation Co-localisation with HBV antigens • Branched amplification	• Tissue: DNA rcDNA cccDNA and integrated DNA DNA and protein DNA/RNA	• HBV DNA localises mostly in the cytoplasm and with two different patterns • localisation in cytoplasmic • nuclear localisation • Transcription from integrated HBV DNA was shown in HBcAg-negative/ HBsAg-positive cells after hybridization	• Targeting of various genes or biomarkers • Detection of single molecules and two to four target sequences in parallel Higher specificity by using several probe sets	• Deproteinization Mechanical/thermal manipulation • Fixed sample • Low resolution • Differences of sensitivity between antibodies	[31,32,33,34,35,36] [37,38] [39]
	Fluorescent probes: • FISH -adding capsids -digitonin permeabilized cells adding capsids	• Single cell analysis: DNA/RNA	• HBV genome liberation at nuclear envelope or inside of nucleus • Capsids need nuclear import factors for interacting with nuclear pore complex, and genome release occurs in inner face	• More resolution than autoradiography or enzymatic probes • Can target Targeting of various genes or biomarkers	• No use in living cells. Does not allow kinetic studies of viral genome	[40] [40,41]
	• Fluorophore-conjugated readout oligos			• Signal amplification through the binding of two oligos targeting the viral genome in complex with two fluorophores.		[42]
	• bDNA fluorescent		• Distribution and quantification of HBV nucleic acids in the cells	• Signal amplification up to 8000-fold thanks to the branched amplification technique (Figure 2) • More efficient detection than FISH	• Deproteinization	
	• RNAscope			• Less time-consuming than ViewRNA ISH		[43,44,45]
	• ViewRNA ISH				• Time-consuming	[46,47,48,49,50,51]

**Table 3 viruses-14-00557-t003:** Candidate techniques for imaging of HBV nucleic acids: Single cell detection.

Technique	Modality	Target	Possible Finding	Advantages	Drawbacks	References
CRISPR/ Cas	• dCas9-tag: -FP -Sun-tag: -tripartite -Sunspot • sgRNA modification -Aptamers: MS2-MCP, PP7 loops • Cas9 active: -STRIDE	DNA/ RNA dsDNA/ ssDNA	• Single molecule detection • HBV nucleic acids kinetic • Quantification • Distinction between cccDNA and partial DNA	• Fixed and living cells • Allows strategies to improve signal/noise ratio and amplification of signal. • Detection of single molecule and low-abundance of nucleic acids, e.g., Sunspot system • No engineering virus • Specificity • Signal amplification	• Needs signal amplification strategies like Sun-tag system, however large complexes may interfere with nucleic acids kinetic • Requires expression of specific factors in the host cells. • Not applicable in living cells • Antibodies specificity	[52] [53] [23]
Non-CRISPR/ Cas	• Aptamer-protein systems	RNA	• Distinction between transcripts (if combined with other methods)	• Fixed and living cells	• Virus engineering • High background • Sensitive folding conditions • Weak photostability	[54]
	• Light-up aptamer-dye systems	RNA	• Single-molecules detection possible • Distinction between transcripts (if combined with other methods)	• Fixed and living cells	• Virus engineering • Sensitive folding conditions • Weak photostability	[55,56]
	• Molecular beacons	RNA, ssDNA	• Single-molecules detection with advanced microscopes • Distinction between transcripts (if combined with other methods)	• Fixed and living cells	• Prone to nucleases • Difficulties with cell entry • Weak photostability • Prone to non-specific signal	[57,58]
	• Quantum-dots / quantum dots-nanobeacons	RNA, ssDNA	• Single-molecule detection • Distinction between transcripts (if combined with other methods)	• Fixed and living cells • Strong signal • Strong photostability	• Cytotoxicity • Difficulties with cell entry due to size • Single-molecules detection	[59,60]
	• ANCHOR	dsDNA	• Single-molecule detection • HBV nucleic acids kinetics possible due to photostability • Distinction between cccDNA and rcDNA	• High contrast • Signal amplification • Living cells • Photostability due to bleached molecules replenishment	• Virus engineering • Effect of OR protein on cells unknown • High cytosolic background	
	• PNA-based probes	RNA, ssDNA/dsDNA	• Distinction between cccDNA and rcDNA (if combined with other methods)	• Fixed and living cells • High sequence specificity and affinity • Resistant to nucleases • Multiple binding modes make them flexible in function	• Difficulties with cell entry • Prone to non-specific signal to hydrophobic surfaces • Binding mode dependent on careful design	[61] [62] [63] [64]
	• Metabolic labeling	dsDNA	• Single-molecules detection • Distinction cccDNA/ rcDNA (if combined with other methods)	• Labelled infectious virus	• Fixed cells only	[65]

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
