# Peer review of "Imaging of Hepatitis B Virus Nucleic Acids: Current Advances and Challenges"

_viruses, 2022, doi:10.3390/v14030557_

Round 1

Reviewer 1 Report

The review article is very interesting and comprehensive, since it addresses an important topic in the HBV field: the lack of reliable technologies enabling visualization and localization of the HBV template, the cccDNA minichromosome, in relation to other HBV markers that are also present within the infected cell. Technological progress in this field would provide valuable information regarding the spatial distribution and loads of the cccDNA in different experimental systems and pathological conditions in patients.  Here, the authors present both main DNA and RNA hybridization technologies that have been used in the past and in some recent studies and also propose newer technologies that could be applied, in principle, for HBV. Main technologies presented are here also supported by various clear figures (cartoons).

As explained below, my main general remark is that the authors focus a lot on issues related to the sensitivity of the different assays (which was certainly a problem in the past, but less for the most advanced hybridization technologies currently available). However, the problem of ensuring the specificity of the cccDNA measurements could be further elaborated. In my experience, the real main challenge for specific cccDNA visualization is the coexistence of very similar HBV DNA forms. This aspect is mentioned mainly in the introduction, but I would strongly recommend to address this topic further when describing previous reports and novel methodologies to enhance the quality of this interesting review. The different difficulties in visualizing HBV RNAs or rcDNA or cccDNA should be stratified better. Some sentences need to be checked and modified for clarity. Orthographic revision is also recommended.

Specific comments

Abstract: "Imaging of the infections in cells is thus a particular challenge as the viral DNA molecules but also HBV mRNAs exist only in a few copies". There is a low number of cccDNA molecules, bur not of HBV RNA and replicative intermediates in HBV replicating hepatocytes, unless special models (or cirrhotic tissues?) are used. This sentence is misguiding and needs to be modified.

Ln 45-47: This authors suggest that HBV can survive cell division. This sentence is however not supported by references. The authors could rather mention studies indicating the opposite, which show reduction of cccDNA loads by cell division (i.e. Allweiss, Gut 2017).

Ln.105-107: "the existing data support that HBV RNA detection by visualizing techniques is equally challenging than that of cccDNA, although...". I do not agree. RNA in situ hybridization is not as challenging as cccDNA visualization if methods as RNAscope or ViewRNA technologies are used. Difficulties may exist only when expression levels are particularly low. How often this may be the case, is currently unclear and need further investigation. Please, revise this sentence.

Ln.117-119: “This might be of importance for distinguishing between the RNA pregenome, which promotes infection, and HBeAg mRNA”. Instead of HBeAg mRNA, I suggest using the correct terminology “precore mRNA”. As further factor adding complexity, they could mention HBV genotype diversity, which may also limit HBV RNA detection in patient samples.

Ln.170-172:  HBV DNA integrations expressing HBsAg can explain why HBcAg-negative cells may be positive for other HBV markers. Yet, it should be also noted that the sensitivity of HBcAg staining may be inferior to that of HBsAg and to in situ Hybridization. Specificity and sensitivity of distinct antibodies may therefore also contribute to the discrepancies described in those reports.

Fig.2 and Ln 182-186: It could be added that several probe pairs are used thus providing robustness against partial target accessibility or degradation. And also that single molecules can be visualized.

Ln232-234: this finding is somehow contradictory and demonstrate that limitations concern also the limited antibodies sensitivity; not only that of the probes. HBcAg should be present also in cytoplasm in cells actively replicating HBV (hence HBV DNA positive). HBcAg staining becomes positive in cytoplasm only when replication levels are high, thus pointing out to the poor sensitivity of most HBc-antibodies available. As mentioned above, the authors should discuss also such limitations and explain the likely reason of this apparent exclusivity (also due to technical limitations of the antibodies).

Ln246-248: do the authors refer here to the ability to detect cccDNA? I guess so, since rcDNA and HBV RNA are highly abundant at least in infected hepatocytes but also upon transfection of hepatoma cells. Please specify.

Pg.7 (Ln.281-291): the differences in cccDNA amounts between HBV and DHBV are expected and in line with other studies. There is however one important viral DNA form which has not yet mentioned in this manuscript and that may also impact visualization and cccDNA measurements and so influence results reported. This is the presence of the so-called protein-free HBV DNA, which should also be present in the cell nuclei. Although this molecule is mostly detected by Southern blotting after Hirt extraction, its existence and likely detection by FISH should be discussed by the authors at some point.

Ln 294-300: The very elegant study of Yue et al. used ViewRNA ISH technologies to visualize HBV DNA and RNA molecules. They also combined the assay with proteins staining. For the comprehensiveness of the review, the author could mention the assay used in that study and also the existence of the similar RNAscope technology, which has been also successfully used to detect HBV RNA alone and in combination with protein staining in liver tissues (i.e. ref.48 in this manuscript). Such technology is equally sensitive (single molecules can theoretically be visualized) but its application less time consuming than ViewRNA ISH (see user instructions).

Ln300-305: As mentioned above, part of these “cccDNA-like” signals determined in the early phases of infection may be due to the presence of incoming virions becoming nuclear PF-rcDNA, not real mature cccDNA. Such scenario would explain the low RNA levels detected at these early times.

Being this a review article, I would strongly encourage the authors to include some considerations about open questions and limitations when summarizing previous reports.

Figure 8: The figure may be correct since these probes are particular. However, I am not sure whether the secondary probe is really located correctly since it suggests hybridization of 2 probes on the same single DNA strain, while the other strain remains free... should not the second probe be on the D-loop? If not, please clarify and point out this aspect more clearly.

Pg. 9 to 11: The authors present and suggest very elegant new technologies for novel applications in the HBV field. Some advantages and drawbacks of the different approaches are summarized. Yet, it seems that the drawbacks may differ both in terms of sensitivity and specificity of the signals depending also on the infection systems used (tissues, cell cultures, live cell imaging). The model aspect could be pointed out as well. Because of the paucity of solid data, potential pros and cons could be expanded in relation to HBV related challenges.

I would suggest writing a short conclusion (take home message) at the end since the article describes a broad range of complex technologies aiming at detecting the different HBV RNA and DNA molecules. 

Minor comments

Abstract: please correct to show the exponential value or use forms such as 10 billions or similar.

Ln 35: I suggest modifying and inverting the two: ...reduce disease progression and HCC risk albeit only after several years….

Ln40: add "in healthy conditions" or similar since the half-life of the hepatocytes may become substantially shorter in the context of liver inflammation / immune responses.

Ln53-54: “clearance of chronic HBV infections under treatment are not better than without”. This sentence may be too "plain" and also not clear as formulated. Consider mentioning reports measuring cccDNA reductions in treated patients but no clearance.

Ln. 80-81: By mentioning potential interferences between cccDNA and integrations, the authors could mention "Although often rearranged and fragmented, this DNA ...".

Ln.82: “The obstacles in visualization specific HBV DNA molecules…” in visualizing ... Please modify.

Ln91: since digital PCR has also limitations (the problem of selective cccDNA measurements remain), the authors should write "which indicated large variations"

Ln.150-151: „Further specificity controls may be based on scrambled sequence, or cells in which the target sequences are absent”. I would also add “or probes detecting unrelated sequences”.

Ln. 208: write mutually instead of mutual

Ln 217-219: According to the recent literature, HBcrAg rather than HBeAg is considered a potential biomarker of cccDNA activity. Or specify “ in HBeAg-positive patients”.

Ln 246: delete “in”

Ln 334: please mention that sgRNA (and add abbreviation) is a combination of the cr and tracrRNAs. Might be confusing otherwise

Ln 338-339: this sentence is confusing. “However, visualisation of HBV DNA molecules by has not been published yet but might be possible. Please rephrase.

The part author contributions, funding etc. still needs to be completed.

Author Response

We appreciated the comments of the reviewers and followed their advices. We added three tables showing synopses of “Problems and possible solutions in HBV nucleic acid detection”, “Advantages and disadvantages of techniques for HBV nucleic acid detection” and “Candidate techniques for imaging of HBV nucleic acids”; the latter focussed on single cell detection. The language was revised.

Please find below our point-to-point answers to the comments.

Reviewer #1

The review article is very interesting and comprehensive, since it addresses an important topic in the HBV field: the lack of reliable technologies enabling visualization and localization of the HBV template, the cccDNA minichromosome, in relation to other HBV markers that are also present within the infected cell. Technological progress in this field would provide valuable information regarding the spatial distribution and loads of the cccDNA in different experimental systems and pathological conditions in patients.  Here, the authors present both main DNA and RNA hybridization technologies that have been used in the past and in some recent studies and also propose newer technologies that could be applied, in principle, for HBV. Main technologies presented are here also supported by various clear figures (cartoons).

As explained below, my main general remark is that the authors focus a lot on issues related to the sensitivity of the different assays (which was certainly a problem in the past, but less for the most advanced hybridization technologies currently available). However, the problem of ensuring the specificity of the cccDNA measurements could be further elaborated. In my experience, the real main challenge for specific cccDNA visualization is the coexistence of very similar HBV DNA forms. This aspect is mentioned mainly in the introduction, but I would strongly recommend to address this topic further when describing previous reports and novel methodologies to enhance the quality of this interesting review. The different difficulties in visualizing HBV RNAs or rcDNA or cccDNA should be stratified better. Some sentences need to be checked and modified for clarity. Orthographic revision is also recommended.

Specific comments

Reviewer #1: Abstract: "Imaging of the infections in cells is thus a particular challenge as the viral DNA molecules but also HBV mRNAs exist only in a few copies". There is a low number of cccDNA molecules, bur not of HBV RNA and replicative intermediates in HBV replicating hepatocytes, unless special models (or cirrhotic tissues?) are used. This sentence is misguiding and needs to be modified.

Answer: Thank you for noticing this. We have changed the sentence to: “Imaging of the infections in cells is thus a particular challenge especially for cccDNA that exists only in a few copies.” (line 19f)

Reviewer #1: Ln 45-47: This authors suggest that HBV can survive cell division. This sentence is however not supported by references. The authors could rather mention studies indicating the opposite, which show reduction of cccDNA loads by cell division (i.e. Allweiss, Gut 2017).

Answer: We have extended the section on cccDNA and cell division substantially (page 2, 2nd para, line 54ff).

Reviewer #1: Ln.105-107: "the existing data support that HBV RNA detection by visualizing techniques is equally challenging than that of cccDNA, although...". I do not agree. RNA in situ hybridization is not as challenging as cccDNA visualization if methods as RNAscope or ViewRNA technologies are used. Difficulties may exist only when expression levels are particularly low. How often this may be the case, is currently unclear and need further investigation. Please, revise this sentence.

Answer: We have modified the sentence which now reads: “HBV RNA detection by visualisation techniques is challenging, but in contrast to cccDNA it does not require the melting of double-stranded structures (see ISH).” (line 121f)

Reviewer #1: Ln.117-119: “This might be of importance for distinguishing between the RNA pregenome, which promotes infection, and HBeAg mRNA”. Instead of HBeAg mRNA, I suggest using the correct terminology “precore mRNA”. As further factor adding complexity, they could mention HBV genotype diversity, which may also limit HBV RNA detection in patient samples.

Answer: We have changed the sentence to: “This might be of importance for distinguishing between the pgRNA which promotes infection, and precore mRNA which encodes HBeAg and is dispensable for infection and propagation process.” (line 126f)

Reviewer #1: Ln.170-172:  HBV DNA integrations expressing HBsAg can explain why HBcAg-negative cells may be positive for other HBV markers. Yet, it should be also noted that the sensitivity of HBcAg staining may be inferior to that of HBsAg and to in situ Hybridization. Specificity and sensitivity of distinct antibodies may therefore also contribute to the discrepancies described in those reports.

Answer: We have changed the sentence to: “In nowadays light, this finding indicates transcription from integrated HBV DNA, as classical integrates are linearized between core promotor and core ORF [22] , even though differences in the sensitivity between antibodies towards HBcAg and HBsAg could also be a contributing factor.” (line 189ff)

Reviewer #1: Fig.2 and Ln 182-186: It could be added that several probe pairs are used thus providing robustness against partial target accessibility or degradation. And also that single molecules can be visualized.

Answer: The sentence has been changed to: “Labelled probes binding to the extensions of the amplifier are then added in a third step (Fig. 2D), which allows amplification by 8000-fold, compared to ISH using a direct single probe. The method thus allows detection of single molecules. Different extensions on the primary oligonucleotides allow detection of two to four target sequences in parallel. Thus, several probe sets may be use to allow specific detection of nucleic acids with only partial accessibility or degradations.” (line 203ff)

Reviewer #1: Ln232-234: this finding is somehow contradictory and demonstrate that limitations concern also the limited antibodies sensitivity; not only that of the probes. HBcAg should be present also in cytoplasm in cells actively replicating HBV (hence HBV DNA positive). HBcAg staining becomes positive in cytoplasm only when replication levels are high, thus pointing out to the poor sensitivity of most HBc-antibodies available. As mentioned above, the authors should discuss also such limitations and explain the likely reason of this apparent exclusivity (also due to technical limitations of the antibodies).

Answer: We decided to remove this sentence as the topic is beyond the scope of the article and the localization of HBcAg is in fact far more complex. E.g. the reviewer stated that “HBcAg staining becomes positive in cytoplasm only when replication levels are high”, which is mostly right but not always. Well described e.g. by Kim and colleagues (J Korean Med Sci, 2006, 21(2):279-83), is the observation that in “…young patients with chronic B viral hepatitis, the degree of expression of HBcAg in the hepatocyte nucleus may affect viral load, and the degree of expression of HBcAg in the hepatocyte cytoplasm may affect histologic activities of liver disease…”. Furthermore, we find the interpretation/statement of the reviewer about the “…poor sensitivity of most HBc-antibodies available…” too apodictic for using it as argument in a discussion. The authors do not and cannot know the sensitivity of all commercial antibodies, which were available at the time of the individual publication. Own experiments, showed a sensitivity of e.g. the polyclonal rabbit antibody against folded HBcAg in the low pictogram amount per c. 10 mm^2 band (enhanced chemo luminescence detection) indicate in fact an extraordinary sensitivity of at least some antibodies.

Reviewer #1: Ln246-248: do the authors refer here to the ability to detect cccDNA? I guess so, since rcDNA and HBV RNA are highly abundant at least in infected hepatocytes but also upon transfection of hepatoma cells. Please specify.

Answer: We have clarified this by changing the sentence to: “However, as capsid-lipofection of cells yields hundreds of intranuclear HBV genome copies, this technique did not require high sensitivity as would be required for HBV cccDNA and rcDNA detection upon natural infection.”. (line 265ff)

Reviewer #1: Pg.7 (Ln.281-291): the differences in cccDNA amounts between HBV and DHBV are expected and in line with other studies. There is however one important viral DNA form which has not yet mentioned in this manuscript and that may also impact visualization and cccDNA measurements and so influence results reported. This is the presence of the so-called protein-free HBV DNA, which should also be present in the cell nuclei. Although this molecule is mostly detected by Southern blotting after Hirt extraction, its existence and likely detection by FISH should be discussed by the authors at some point.

Answer: Thank you for this suggestion. We have included a discussion about protein free rcDNA: “Noteworthy, the assay was not specific for cccDNA, but would also detect rcDNA from viral capsids that had entered the nucleus that is converted to cccDNA through protein free rcDNA”. (line 310ff)

Reviewer #1: Ln 294-300: The very elegant study of Yue et al. used ViewRNA ISH technologies to visualize HBV DNA and RNA molecules. They also combined the assay with proteins staining. For the comprehensiveness of the review, the author could mention the assay used in that study and also the existence of the similar RNAscope technology, which has been also successfully used to detect HBV RNA alone and in combination with protein staining in liver tissues (i.e. ref.48 in this manuscript). Such technology is equally sensitive (single molecules can theoretically be visualized) but its application less time consuming than ViewRNA ISH (see user instructions).

Answer: Thank you for noticing this. We have specified whether ViewRNA ISH or RNAscope was used in the bDNA studies and in the new table 2 describing the techniques.

Reviewer #1: Ln300-305: As mentioned above, part of these “cccDNA-like” signals determined in the early phases of infection may be due to the presence of incoming virions becoming nuclear PF-rcDNA, not real mature cccDNA. Such scenario would explain the low RNA levels detected at these early times.

Answer: We have clarified that (-) strand HBV DNA and not specifically cccDNA was assayed.

Reviewer #1: Being this a review article, I would strongly encourage the authors to include some considerations about open questions and limitations when summarizing previous reports.

Answer: We have extended the summary paragraph in the end of section 3 to include a discussion of limitations and open questions. We have also extended the discussion of limitations in the Yue et al reference. We further added table 1 in which the problems in detection and potential solutions are described.

Reviewer #1: Figure 8: The figure may be correct since these probes are particular. However, I am not sure whether the secondary probe is really located correctly since it suggests hybridization of 2 probes on the same single DNA strain, while the other strain remains free... should not the second probe be on the D-loop? If not, please clarify and point out this aspect more clearly.

Answer:  We removed figure 8D to improve clarity.

Reviewer #1: Pg. 9 to 11: The authors present and suggest very elegant new technologies for novel applications in the HBV field. Some advantages and drawbacks of the different approaches are summarized. Yet, it seems that the drawbacks may differ both in terms of sensitivity and specificity of the signals depending also on the infection systems used (tissues, cell cultures, live cell imaging). The model aspect could be pointed out as well. Because of the paucity of solid data, potential pros and cons could be expanded in relation to HBV related challenges.

Answer: We made it clearer in the body of text that it is within the context of cell culture. Also, a discussion section and a table with pros and cons have been added.

Reviewer #1: I would suggest writing a short conclusion (take home message) at the end since the article describes a broad range of complex technologies aiming at detecting the different HBV RNA and DNA molecules. 

Answer: Thank you for this suggestion. We have added a discussion section which –in combination with the tables – gives a short overview about the achievements, the obstacles and possible solutions.

Minor comments

Reviewer #1: Abstract: please correct to show the exponential value or use forms such as 10 billion or similar.

Answer: This has been corrected. (line 17)

Reviewer #1: Ln 35: I suggest modifying and inverting the two: ...reduce disease progression and HCC risk albeit only after several years….

Answer: This has been corrected. (line 35f)

Reviewer #1: Ln40: add "in healthy conditions" or similar since the half-life of the hepatocytes may become substantially shorter in the context of liver inflammation / immune responses.

Answer: This has been corrected as suggested. (line 40)

Reviewer #1: Ln53-54: “clearance of chronic HBV infections under treatment are not better than without”. This sentence may be too "plain" and also not clear as formulated. Consider mentioning reports measuring cccDNA reductions in treated patients but no clearance.

Answer: Thank you for this important comment. We modified the phrase in that we added how clearance is defined. This is important as a significant number of chronic HBV infections relapse upon immune suppression, even when anti-HBs was present. “Consequently, clearance of chronic HBV infections as indicated by the loss of HBsAg under treatment are not better than without [8] [9], and therapeutic interventions targeting cccDNA represent the “holy grail”.” (line 51ff)

Reviewer #1: Ln. 80-81: By mentioning potential interferences between cccDNA and integrations, the authors could mention "Although often rearranged and fragmented, this DNA ...".

Answer: This has been corrected. (line 89f)

Reviewer #1: Ln.82: “The obstacles in visualization specific HBV DNA molecules…” in visualizing ... Please modify.

Answer: This has been corrected. (line 101ff)

Reviewer #1: Ln91: since digital PCR has also limitations (the problem of selective cccDNA measurements remain), the authors should write "which indicated large variations"

Answer: This has been corrected. (line 110)

Reviewer #1: Ln.150-151: „Further specificity controls may be based on scrambled sequence, or cells in which the target sequences are absent”. I would also add “or probes detecting unrelated sequences”.

Answer: This information has been added. (line 168)

Reviewer #1: Ln. 208: write mutually instead of mutual

Answer: This has been corrected. (line 168)

Reviewer #1: Ln 217-219: According to the recent literature, HBcrAg rather than HBeAg is considered a potential biomarker of cccDNA activity. Or specify “ in HBeAg-positive patients”.

Answer: “in HBeAg-positive patients” has been added. (line 240)

Reviewer #1: Ln 246: delete “in”

Answer: This has been corrected. (line 265)

Reviewer #1: Ln 334: please mention that sgRNA (and add abbreviation) is a combination of the cr and tracrRNAs. Might be confusing otherwise

Answer: The following sentence has been added:” Single-guide RNA (sgRNA) combines the crRNA and tracrRNA into a single RNA molecule. crRNA contains 17-20 nucleotides complementary to the target DNA while tracrRNA serves as scaffold for the Cas nuclease.” (line 368ff)

Reviewer #1: Ln 338-339: this sentence is confusing. “However, visualisation of HBV DNA molecules by has not been published yet but might be possible. Please rephrase.

Answer: The sentence has been changed to “However, imaging of HBV nuclei acids by CRSPR/Cas has not been published yet but might be promising.” (line 375f)

Reviewer #1: The part author contributions, funding etc. still needs to be completed.

Answer: We have added information about funding and author contribution.

Reviewer 2 Report

The review is nice, it is well written, comprehensive. The first half is about the techniques that have been applied to detect HBV DNA (and a bit on RNA), the second one lists a random number of random techniques to detect DNA. An explanation why o how these techniques should be applied in HBV DNA detection is often lacking. The figures are really nice and help with the understanding of the text. Some minor spelling mistakes were observed, but nothing interfering with understanding.

Major comments:

The title doesn’t give the reader any idea of what the review is about. Also, there is a strong focus on dsDNA detection. The problems of detecting the different HBV transcripts, which is difficult due to their overlap, is underdiscussed and no techniques applied to overcome this problem are reviewed. For instance, for attempts to detect the different HBV transcripts see: doi:10.1128/JVI.01625-16 , https://pubmed.ncbi.nlm.nih.gov/8077209/ , https://pubmed.ncbi.nlm.nih.gov/32087349/

Also, no review strategy is provided.

Many complicated methods to detect DNA are briefly touched upon without much discussion whether these have any potential to outperform previously applied techniques. Propper discussion and weighing the pro’s and cons of the methods may substantially improve the review? Or for instance adding a table with the different techniques, their pro’s and con’s, and whether they have been applied and where (references). What is really lacking is the more “classic” dsDNA detecting proteins, the Zinc-finger nucleases and the TALEN proteins, and the proximity ligation assay (although variants are discussed). These should be added.

Minor comments:

The remark about the life span of hepatocytes (line 40) is confusing and inconsistent with the reference; the article referenced by the authors (!!) states “Under normal conditions, the hepatocyte population is self-renewing, with most hepatocyte able to divide to maintain liver mass. The hepatocyte turnover rate is uncertain.”. Referencing a more recent paper on hepatocyte proliferation/turnover may be more appropriate.

Related to this, the notion that dividing hepatocytes may pass cccDNA to daughter cells is too brief for such an important point and the authors should describe and reference this in more detail.

Number of infected hepatocytes (lines 48-50: Please expand with HBV expressing hepatocytes in humans, in chronic infection and under treatment)

Lines 171-172: In HBV infected liver, there seem to be both only-core and only-HBsAg positive cells. As far s I know, this may also be due to expression patterns from the cccDNA. So please include this notion or strengthen your current reasoning with appropriate references. It would be interesting to measure the transcriptional basis for this phenomenon, based on the title I kind of hoped to find this in this review but alas it seems that the authors ignore this notion.

Chapter 2 is about in situ DNA (and nothing on RNA) detection

Lines 208-209: a “mutual exclusive pattern between HBsAg and cccDNA”? I don’t understand what the authors mean by this

Line 254: “HBV-infected HepG2-NTCP and HepAD38 cell lines” The HepAD38 cell line has integrated HBV DNA, i.e., it replicates HBV but is not infected. This difference between the models should be taken in account when evaluating the merits of this technique in this study.

Line 308: “The ratio” ratio of what to what?

Line 364: Check font size. Also the used antibody is a single chain variable fragment specific for the tag I assume?

Fig. 3 legend, the SUN tag is recognised by a fluorescent probe, and is not fluorescent by itself. Adding the PAMmer to the box would make it a bit clearer

Lines 415-427: Check font size

The paragraph on aptameres, and lines 554-556: This makes me wonder whether the epsilon loop in the HBV pgRNA could be used as a “natural” aptamer?

Author Response

We appreciated the comments of the reviewers and followed their advices. We added three tables showing synopses of “Problems and possible solutions in HBV nucleic acid detection”, “Advantages and disadvantages of techniques for HBV nucleic acid detection” and “Candidate techniques for imaging of HBV nucleic acids”; the latter focussed on single cell detection. The language was revised.

Please find below our point-to-point answers to the comments.

Reviewer #2

The review is nice, it is well written, comprehensive. The first half is about the techniques that have been applied to detect HBV DNA (and a bit on RNA), the second one lists a random number of random techniques to detect DNA. An explanation why or how these techniques should be applied in HBV DNA detection is often lacking. The figures are really nice and help with the understanding of the text. Some minor spelling mistakes were observed, but nothing interfering with understanding.  

Major comments:

Reviewer #2: The title doesn’t give the reader any idea of what the review is about. Also, there is a strong focus on dsDNA detection.

Answer: The title was suggested by the editor. Nonetheless, we changed it to: “Imaging of hepatitis B virus nucleic acids: current advances and challenges”

Reviewer #2: The problems of detecting the different HBV transcripts, which is difficult due to their overlap, is underdiscussed and no techniques applied to overcome this problem are reviewed. For instance, for attempts to detect the different HBV transcripts see: doi:10.1128/JVI.01625-16 , https://pubmed.ncbi.nlm.nih.gov/8077209/

Answer: We agree that the problem of specifically detecting different HBV transcripts is important and have included a discussion section, which includes the interesting references suggested by the reviewer and another study. We further added table 1 in which these problems are pointed out.

We have also included a description on how Yue et al. addressed this problem.

Reviewer #2: Also, no review strategy is provided.

Answer: To describe the review strategy, we have included a description of the aim of the review in the end of the introduction (section 1):

“The aim of this review is to provide an overview of the challenges of HBV genome- and transcript-detection (Table 1), and how microscopical studies has increased the understanding of HBV infection (Table 2). Furthermore, recent approaches which have not been applied to HBV research are described and their potential for single-molecule detection and tracking of HBV nucleic acids are discussed (Table 3).” (line 134ff)

Reviewer #2: Many complicated methods to detect DNA are briefly touched upon without much discussion whether these have any potential to outperform previously applied techniques. Propper discussion and weighing the pro’s and cons of the methods may substantially improve the review? Or for instance adding a table with the different techniques, their pro’s and con’s, and whether they have been applied and where (references). What is really lacking is the more “classic” dsDNA detecting proteins, the Zinc-finger nucleases and the TALEN proteins, and the proximity ligation assay (although variants are discussed). These should be added.

Answer: A discussion section and table with pro´s and con´s have been added.

Future developments section focused on methods previously applied to imaging of viral nucleic acids. However, zinc-finger and TALEN briefly addressed at the end of section 4 indicating there are more methods which have not yet discussed.

Minor comments:

Reviewer #2: The remark about the life span of hepatocytes (line 40) is confusing and inconsistent with the reference; the article referenced by the authors (!!) states “Under normal conditions, the hepatocyte population is self-renewing, with most hepatocyte able to divide to maintain liver mass. The hepatocyte turnover rate is uncertain.”. Referencing a more recent paper on hepatocyte proliferation/turnover may be more appropriate.

Answer: Thank you for noticing this. We have included a more recent review and changed the sentence to: HBV has a partially double-stranded DNA genome (relaxed circular; rc), and infects hepatocytes which in healthy conditions have a long life-span (around 200-400 days in rodents)

Reviewer #2: Related to this, the notion that dividing hepatocytes may pass cccDNA to daughter cells is too brief for such an important point and the authors should describe and reference this in more detail.

Answer: We have extended the section on cccDNA and cell division and included several additional references.

Reviewer #2: Number of infected hepatocytes (line 48-50: Please expand with HBV expressing hepatocytes in humans, in chronic infection and under treatment)

Answer: We added the information that these numbers were based on “immune histology detecting the HBV core protein in chimpanzees with acute HBV infection [6] [7]”. (line 47f).

Reviewer #2: Line 171-172: In HBV infected liver, there seem to be both only-core and only-HBsAg positive cells. As far as I know, this may also be due to expression patterns from the cccDNA. So please include this notion or strengthen your current reasoning with appropriate references. It would be interesting to measure the transcriptional basis for this phenomenon, based on the title I kind of hoped to find this in this review but alas it seems that the authors ignore this notion.

Answer: We have included a discussion on alternative explanation to this observation: It has been suggested that HBsAg regulates cccDNA levels with high levels of HBsAg production blocks cccDNA synthesis” (line 192f)

Reviewer #2: Chapter 2 is about in situ DNA (and nothing on RNA) detection

Answer: We have changed the title to “Techniques applied for HBV nucleic acid visualisation”

Reviewer #2: Line 208-209: a “mutual exclusive pattern between HBsAg and cccDNA”? I don’t understand what the authors mean by this

Answer: This sentence was commented also by reviewer 1 and we decided to remove it since the reference is discussed also in section 3.1

Reviewer #2: Line 254: “HBV-infected HepG2-NTCP and HepAD38 cell line” The HepAD38 cell line has integrated HBV DNA, i.e., it replicates HBV but is not infected. This difference between the models should be taken in account when evaluating the merits of this technique in this study.

Answer: We have clarified that HepAD38 cells are transfected with cDNA corresponding to pgRNA and added that the higher signal observed in those cells compared to infected HepG2-NTCP cells is expected. (line 273ff)

Reviewer #2: Line 308: “The ratio” ratio of what to what?

Answer: We have clarified this by changing the sentence to: Adding entecavir, which blocks reverse transcription, reduced the number of DNA molecules but not pgRNA. Further, treatment with Interferon α reduced both DNA and RNA, while inhibitors of capsid assembly reduced the number of pgRNA puncta co-localised with capsid by 48-57%.” (line 331ff)

Reviewer #2: Line 364: Check font size. Also the used antibody is a single chain variable fragment specific for the tag I assume?

Answer: This has been corrected. The sentence has been rephrased to make it easier understand by The Sun-Tag system is based on a scaffold protein that contains 24 copies of short epitopes GCN4 (General Control protein 4), which is recognised by the fused to GFP”. (line 400ff)

Reviewer #2: Fig. 3 legend, the SUN tag is recognised by a fluorescent probe, and is not fluorescent by itself.

Answer: Thank you for noticing this. we have changed the sentence to:dCas9 is fused to a single fluorescent protein or conjugated to multiple fluorescent proteins through Sun-Tag”. (line 415f)

Reviewer #2: Adding the PAMmer to the box would make it a bit clearer

Answer: This has been done.

Reviewer #2: Line 415-427: Check font size

Answer: This has been corrected.

Reviewer #2: The paragraph on aptameres, and line 554-556: This makes me wonder whether the epsilon loop in the HBV pgRNA could be used as a “natural” aptamer?

Answer: Thank you for the suggestion. A sentence was added at the end of section 4 mentioning that the epsilon could potentially be one of secondary structures targeted for imaging: “An attractive option would be the visualisation of native viral genomes in situ through their structural properties such as the HBV pgRNA epsilon.” (line 600ff)

Reviewer 3 Report

In this review article, Dr. Kann and his team discussed microscopical techniques that have been used, or have the potential to be used, in visualizing HBV genomic DNA (mainly cccDNA, but also includes rcDNA) and mRNA transcripts, the dynamics of which has great implications in understanding HBV infection and treatment outcomes. They first reviewed the significance and technical challenges of visualizing cccDNA and mRNA in HBV infected cells/tissues, and then summarized the major findings of several representative studies using such visualization techniques including electron microscopy, in-situ hybridization, and fluorescent in-situ hybridization (FISH). This summary is comprehensive, including studies from 1980s to the most recent ones. Lastly, they discussed recent developments of several new technologies in nucleic acid detection that had been successfully applied in other areas but not yet in HBV research, such as the CRISPR/Cas system, aptamers, molecular beacons, peptide nucleic acid (PNA)-based probes, and etc.. This summary is insightful and surely will be very useful to researchers in the HBV field and beyond.

I recommend the publication of this review article but would suggest that a professional editing service is needed to improve the language before it can be accepted for publishing.

Author Response

We appreciated the comments of the reviewer. We added three tables showing synopses of “Problems and possible solutions in HBV nucleic acid detection”, “Advantages and disadvantages of techniques for HBV nucleic acid detection” and “Candidate techniques for imaging of HBV nucleic acids”; the latter focussed on single cell detection. The language was revised.

Please find below our point-to-point answers to the comments.

Reviewer 3

In this review article, Dr. Kann and his team discussed microscopical techniques that have been used, or have the potential to be used, in visualizing HBV genomic DNA (mainly cccDNA, but also includes rcDNA) and mRNA transcripts, the dynamics of which has great implications in understanding HBV infection and treatment outcomes. They first reviewed the significance and technical challenges of visualizing cccDNA and mRNA in HBV infected cells/tissues, and then summarized the major findings of several representative studies using such visualization techniques including electron microscopy, in-situ hybridization, and fluorescent in-situ hybridization (FISH). This summary is comprehensive, including studies from 1980s to the most recent ones. Lastly, they discussed recent developments of several new technologies in nucleic acid detection that had been successfully applied in other areas but not yet in HBV research, such as the CRISPR/Cas system, aptamers, molecular beacons, peptide nucleic acid (PNA)-based probes, and etc. This summary is insightful and surely will be very useful to researchers in the HBV field and beyond.

I recommend the publication of this review article but would suggest that a professional editing service is needed to improve the language before it can be accepted for publishing.

Thank you very much. We deeply appreciated your comments. The language has been revised.

Round 2

Reviewer 1 Report

This is an interesting and comprehensive overview which summarizes and explains main technologies that have been used to visualize distinct HBV nucleic acids in different systems. In the extensively revised version, the authors clarified all main concerns raised in the review process. In particularly the problematic related to cccDNA visualization is now clearly explained. Also the new technologies that could be employed in the HBV field have been put now in the right perspective, highlighting potential pros and cons of each technology in general and for HBV research in particular.

Overall and despite the highly technical character of the manuscript, this is now a well written and very comprehensive review article, accompanied by clear drawings and focusing on an important topic in HBV research, which is key to advance our understanding in HBV biology and treatment responses.

I have only very few minor remarks:

  1. 664: … sets may be used to allow…. Please, correct.

Figure 2: the yellow lines are not well visible. Consider change colour.

Ln. 1024-26: suggest rephrasing: “However, a symmetrical passage to daughter cells was not confirmed for HBV, where a loss of cccDNA was rather observed.”